# Vulnerability Analysis of Power Transmission Grids Subject to Cascading Failures

Francesco Cadini [1,*], Luca Lomazzi [1] and Enrico Zio [2,3]

1    Politecnico di Milano, Department of Mechanical Engineering, Via La Masa n.1, 20156 Milan, Italy; luca.lomazzi@polimi.it
2    Politecnico di Milano, Energy Department, Via La Masa n.34, 20156 Milan, Italy; enrico.zio@polimi.it
3    Centre for Research on Risks and Crises, MINES ParisTech, PSL Research University, 06904 Paris, France
*    Correspondence: francesco.cadini@polimi.it

**Abstract:** Cascading failures are a major threat to interconnected systems, such as electrical power transmission networks. Typically, approaches proposed for devising optimized control strategies are demonstrated with reference to a few test systems of reference (IEEE systems). However, this limits the robustness of the proposed strategies with respect to different power grid structures. Recently, this issue has been addressed by considering synthetic networks randomly generated for mimicking power transmission grids' characteristics. These networks can be used for investigating the vulnerability of power networks to cascading failures. In this work, we propose to apply a recent algorithm for sampling random power grid topologies with realistic electrical parameters and further extend it to the random allocation of generation and load. Integration with a realistic cascade simulation tool, then, allows us to perform thorough statistical analyses of power grids with respect to their cascading failure behavior, thus offering a powerful tool for identifying the strengths and weaknesses of different grid classes. New metrics for ranking the control and mitigation effort requirements of individual cascade scenarios and/or of grid configurations are defined and computed. Finally, genetic algorithms are used to identify strategies to improve the robustness of existing power networks.

**Keywords:** random power transmission grids; cascading failures; control and mitigation measures





## 1. Introduction

The reliable and safe operation of power transmission grids is of paramount importance for the prosperity of modern society. Power outages and interruptions in the U.S. have been estimated to cost around 150 billion dollars per year. The major blackout of Northeast America in 2003 resulted, alone, in a USD 6 billion economic loss for the region [1,2]. Moreover, the social consequences of power interruptions, such as those related to transportation, food storage and credit card operations, just to mention a few, are no less serious than economic consequences [3].

Electrical power blackouts are caused by cascading failures, usually initiated by the failure of a limited set of components, often caused, in turn, by external events such as lightning, ice, and other extreme weather conditions. Other components failures and disconnections may then propagate across the transmission network, due to the following power load re-distribution among the still-functioning components, which may pass their design load capacities and fail or be disconnected to avoid further severe damage.

Various research efforts have focused on the risk of cascading failures in power transmission networks, analyzing temporal series of blackout data [4] to build probabilistic distributions by developing simulation models to describe cascade dynamics [5–8] and by defining and calculating quantitative indicators for identifying system vulnerabilities to cascading failures [9,10].

These efforts are also of relevance for the successful development of the new concept of the "smart grid" [3], which involves the evolution of the current centralized power generation structures into distributed ones that are interconnected via a properly designed and operated information and communication network (ICT) for improving its observability and controllability [11]. Indeed, the interconnection of the power network to a fast and reliable ICT network allows, in principle, thorough online monitoring of the system with improved coordination of the control and protection systems; automated control strategies may then promptly detect and isolate small contingencies before they give rise to catastrophic cascading failures, or, at the very least, they may contribute to the effective mitigation of their damage [12–14]. Understanding of the dynamics of power system cascading failures is essential for devising more reliable, safe, and economically sound infrastructure designs to improve mitigation and control strategies and to provide more accurate risk assessments. The complexity of the cascading failure process in an electrical power grid is mainly due to the concurrence of the physics of power flows and discrete stochastic network topology reconfigurations, as components become disconnected from the grid as a result of the failure cascade.

For cascading failure analysis, the power transmission network is typically modeled as a graph $G = (V, E)$, where $V$ is the set of vertices (nodes) representing generators, transmission, and load buses and $E$ is the set of edges (arcs) representing the power lines linking the network components. Usually, it is assumed that only elements belonging to one of the sets $V$ or $E$ are subjected to failure; consequently, the analyses are carried out under the assumptions of node removals [7,15–17] or edge removals [5,6,10,18]. In this work, we restrict our attention to only edge removals, since power transmission line failures are more common than bus failures [19].

Under this graph-based framework, cascading failure models can be relatively simple, conceptually. First, a capacity $C_l$ is a assigned to each edge $l \in$, to represent the maximum amount of power flow $F_l$ that can safely flow through line $l$. Then, a single edge is assumed to fail and it is removed from the network, giving rise to a power redistribution transient according to given rules. During the transient, whenever $l$ $F_l > C_l$ in a line, such a line can fail with probability $p$. When a line fails, it is removed from the set $E$, thus further modifying the topology of the network; then, the power flowing in the network is redistributed again, possibly giving rise to further failures or disconnections.

Typically, two approaches are considered for cascading failure modeling: (i) complex network (abstract) models and (ii) realistic power flow models considering alternate current (AC) or linearized direct current (DC) schemes.

In the first kind of model, such as the Motter–Lai model [7], a generic flow unit is assumed to travel along the shortest paths joining pairs of nodes, without specification and consideration of electrical parameters; extensions of this model can be found, for example, in Ref. [6], where the properties of network connection efficiency are introduced for evaluating the flows; in Ref. [10], where the exceeding loads are propagated on the neighboring components by a simple topological re-dispatch rule; and in Ref. [20], where a random flow model is proposed. Although simple and computationally not expensive, these models could be too far from the real physical behavior of power systems [21]. Nevertheless, they have been applied to analyze a broad range of networks ranging from randomly generated networks to some real power transmission networks [15,17].

The second class of models aims to overcome the limitations of complex network models by considering a more realistic electrical network model whose power flows are governed by Kirchoff laws. Then, a system of nonlinear equations describing the physics of the alternate current (AC) power flow can be derived [22]. Under normal operating conditions, a solution of the system can be obtained by means of the Newton–Raphson method. However, under extremely dynamic operating conditions, such as those arising in cascading failures, the Newton–Rapshon approach may be too slow or may even fail to converge, thus hampering any analysis requiring repeated network model evaluations (e.g., for probabilistic risk assessment, uncertainty propagation, or sensitivity

analysis). For these reasons, often, the linearized version of the AC power flow is preferred, which assumes a direct current (DC) model and other simplifying assumptions [22]. The major problem related to the use of this kind of model is that the electrical data of actual power networks, such as generator and load locations, transformer voltage magnitudes, and line impedances and capacities, are available to the scientific community only for a handful of systems, e.g., IEEE test systems (the Power System Test Archive, UWEE). This fact has limited the application of cascading failure models to a restricted set of case studies with very specific topologies and electrical parameters, significantly hindering the generality of the results obtained and of the conclusions drawn. These limitations should not be underestimated, since many researchers have already pointed out the importance of the topological configuration and electrical properties in the assessment of the network behavior with respect to cascading failure [9,15,23] and voltage stability [24]. Moreover, many of the control and mitigation algorithms presented in the literature [12–14] are developed on the basis of the power flow models discussed above; thus, in general, their performances have been tested on the same restricted set of case studies, leaving many unanswered questions about their robustness and efficacy in different operating contexts.

The first objective of this work is that of overcoming some of the limitations of the models described above by exploiting the recently proposed algorithm for generating random artificial power networks characterized by topological and electrical properties in statistical agreement with those of real power transmission systems [25]. To this end, we first extend the algorithm of Ref. [25] in order to be able to also sample the power supply and demand locations and magnitudes. The extension allows for further enlargement of the space of grid configurations available for statistical analysis and provides a means for possibly capturing the role played by the uncertain distributions of the loads and generators, which are mainly due to the continuous shift towards the increasingly distributed generation schemes of the new-generation power grids [3]. Note that the random networks generated with the algorithm in Ref. [25] do not in general fulfill the $N - 1$ network design basic criterion, which requires that at least two line failures are necessary for a cascading failure event to be triggered [26]. Then, the algorithm is coupled to a model of cascading failures based on a DC power flow approximation and relying on a minimal, proportional re-dispatch control scheme, which maintains the power balance in each network island formed during the cascades. A dynamic power inertia model taken from Ref. [12] is also introduced to realistically account for the temporal evolution of the cascading failures.

The proposed computational model allows us to perform statistical analyses on different families of power grids with respect to their cascading failure behaviors.

An additional original contribution of this work, derived from the computer simulations generated with the tool described above, is the definition of new metrics for ranking the control and mitigation effort requirement of individual accidental scenarios and/or of the power grid configurations with respect to cascading failures. In order to do so, the major cascade consequences—i.e., the final load shedding, the number of lines disconnected due to overloads, and the cascade durations—are properly accounted for in the proposed definitions.

Finally, a genetic algorithm-based procedure aiming to optimize the proposed metrics is proposed in order to identify possible strategies for improving the robustness of an existing power network with respect to cascading failures.

The proposed computational approach is demonstrated with regard to random power transmission networks derived from the IEEE14 and the IEEE118 reference grids (Power System Test Archive, UWEE).

The paper is structured as follows. Section 2 reviews some of the topological properties of the real data available on power transmission networks and briefly explains the RT-nestedSW algorithm of Ref. [25]; Section 3 recalls the DC power flow model, illustrates the details of the cascading failure model employed throughout the whole work, and introduces the standard measures usually employed to quantify cascading failure damage; Section 4 illustrates the simulation results and introduces the new metrics for the cascade

scenarios and the sampled networks; and finally, Section 5 concludes the work and outlines possible future research issues.

## 2. Random Generation of Power Grid Configurations

Previous works on modeling power transmission networks have pointed out the need to generate random power grid test cases of scalable sizes. A few solutions have been proposed for this purpose. For example, Ref. [27] proposed ring-like structures to study the pattern and the velocity of contingency propagation, and Ref. [18] used a tree structure network to study critical points and detect transition points in power system blackouts. However, both types of random network generators neither accurately reflect the topologies of real power transmission grids nor account for the electrical properties of the networks (e.g., loads, generation, and impedances), which together determine the behavior of a power transmission system including the dynamic propagation of oscillations and disturbances in the grid. The recent RT-nested Small World (RT-nestedSW) algorithm introduced by Ref. [25] significantly overcomes both the topological and the electrical deficiencies of previous approaches.

In this section, we recall the main features of the RT-nestedSW algorithm introduced by Ref. [25] for randomly generating plausible realistic power grid topologies, and we also explain the extension we made in order to be able to generate load and generator schemes with random positions and magnitudes. For further details, the interested reader is referred to Ref. [25].

From a purely topological point of view, a power network can be represented by an undirected graph $G = (V, E)$, with $|V| = N$ nodes representing the substations in the system (i.e., generators, loads and transmission nodes) and $|E| = N_L$ links representing the transmission lines interconnecting the substations. Often, the topologies of power grids are considered small-world networks [28]. However, although real power network systems bear some similarities with small-world networks, in Ref. [25], it was shown that they have significantly better connectivity scaling laws. In fact, the average nodal degree $< k >$ is basically constant and does not scale with the network size, as in the case of small-world grids [28]. A detailed analysis of the topological properties of real power transmission networks is out of the scope of this work; for more complete studies of these systems and their relationships with standard random graph models, the interested reader is referred to [25,29,30].

In order to be able to reproduce the random wiring of a power transmission network, Ref. [25] proposed a new approach based on nesting several small-world sub-networks into a regular lattice. By doing so, it is possible to generate networks characterized by a connectivity lying between those of one-dimensional and two-dimensional lattices with better scaling properties than the standard small-world model. The RT-nested SW algorithm proceeds in a hierarchical way for generating random power grids: first, it produces connected sub-networks, and then it interconnects the sub-networks by means of lattice connections. Finally, it generates line impedances by sampling from proper probability distributions estimated on the basis of the available data.

The generation of the sub-networks stems from an algorithm different from the classical one proposed by Ref. [28], with differences lying mainly on the link selection and rewiring procedures. With regard to the link selection, instead of generating links by connecting the most immediate $\frac{<k>}{2}$ neighboring nodes to form a regular lattice, the RT-nested SW algorithm selects a number $k$ (sampled from a geometric distribution with an expected value $< k >$) of links at random from a local neighborhood of $N_{d0}$, with $d_0$ being a properly defined distance threshold. The local neighborhood for node $i$ is defined as the group of nodes with a mutual node index difference less than $d_0$: $N_{d0}^{(i)} = \{j; |j - i| < d_0\}$. With regard to link rewiring, in Ref. [25], the authors exploited the apparent correlation between the rewires for building a Markov chain with transition probabilities $(\alpha, \beta)$ in order to select clusters of nodes and, therefore, groups of links to be rewired. After the Markov chain is run $N$ times, clusters of nodes are obtained, which are labeled "1" if they have to be

rewired or "0" if not. Then, by a specific probability $q_{rw}$, some links are selected to rewire from all the links originating from each "1" cluster of nodes, and the corresponding local links are rewired to outside "1" clusters.

In the second step of the algorithm, lattice connections are sampled from neighboring sub-networks to form the main connected network.

Finally, line impedances are randomly generated from a heavy-tailed distribution properly estimated from the data available. The $N_L$ realizations are then sorted by magnitude in ascending order and grouped into local links, rewire links, and lattice connections, according to the range of values they belong to, as shown in Figure 1. The ranges are properly defined on the basis of physical considerations. The line impedances in each interval are then assigned randomly to the corresponding group of links in the topology.

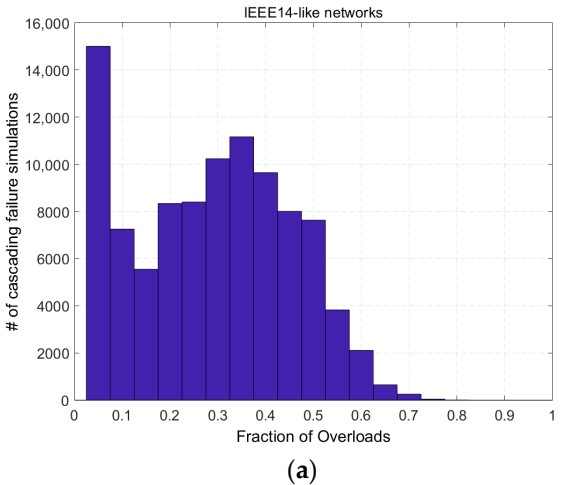

(**a**)

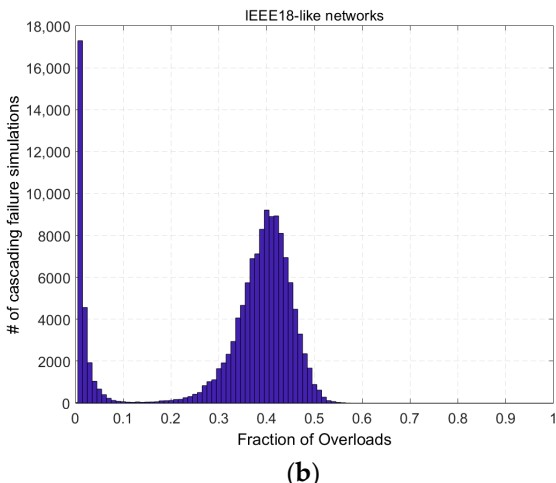

(**b**)

**Figure 1.** Histograms of the fraction of overloads $FO_{G_j,i}$ of the cascading failure scenarios for the IEEE14-like (**a**) and the IEEE118-like (**b**) random networks.

It is important to underline that the data used to estimate the distributions adopted in the above procedure are those related to a single, specific real power transmission network. The RT-nested SW algorithm, then, allows us to generate plausible networks that are statistically similar to the grid chosen as the reference. The work of Ref. [25] also provides the statistical tools needed to extract the required distributions from the reference network.

In particular, in this work, we aim to obtain transmission grids similar to the historical IEEE14 and IEEE118 test cases (from the UWEE Power System Test Case Archive). In order to do so, we fix the number of generators ($N_G$) equal to the number of buses in the IEEE system under consideration, which have non-zero power generation ($PG_i^{IEEE} > 0$). Moreover, the nominal power production vector **PG**, where each component $PG_i$ is the power generated by node $i$ in our network, is a simple random permutation of $\mathbf{PG}^{IEEE}$, the nominal power production vector for the IEEE system under consideration. The total power demand $PD_{tot} = \sum_{i=1}^{NG} PG_i = PG_{tot}$ is, then, randomly allocated among the nodes which have $PG_i = 0$, so that the power balance constraint within the network is satisfied, in agreement with the DC power flow modeling of the system [22]:

$$\sum_{i=1}^{N_G} PG_i = \sum_{j=1}^{N_D} PD_j \tag{1}$$

The portion of $PD_{tot}$ to be assigned to node $j$ with $PG_j = 0$ is sampled from a symmetric Dirichlet distribution in the following way:

$$\frac{PD_j}{PD_{tot}} \sim D(\mathbf{1})$$

where **1** is a $ND = N - NG$ dimensional vector of all ones.

The maximum power that each generator can produce is assumed to be a fraction $\rho$ larger than nominal:

$$PMAX_i = PG_i * (1 + \rho) \quad i = 1, 2 \ldots N_G \tag{2}$$

where $PG_i$ is the power produced by generator $i$. The line capacities $C_l$, $l \in E$ are assigned by the following procedure:

1. The power produced by each generator is assumed to be at its maximum: $PG_i = PMAX_i$ for each $i = 1, 2, \ldots, N_G$; accordingly, $PD_j$ is increased by the same fraction for each $j = 1, 2, \ldots, N_D$ in order for (1) to be satisfied.
2. The corresponding power $F_l$ flowing in each line $l \in E$ of the transmission network is computed by means of a DC power flow model with no losses, solved by the Matlab function MATPOWER [31].
3. The line capacities $C_l$ are then assigned as follows:

$$C_l = F_l(1 + \varepsilon) \quad \forall l \in E \tag{3}$$

where $\varepsilon$ is a parameter playing the same role of $\rho$ for the power generation. Alternatively, different strategies for more realistically assigning both the maximum generator power and the line capacities could in principle be followed, based, for example, on sampling values of both $\rho$ and $\varepsilon$ from properly devised probability distributions or on calculating them on the basis of physical laws and/or engineering practices. However, we believe this is outside the scope of the present work, which is mainly methodological, and that the assumptions made are sufficient to capture the average properties of cascading failures in power transmission networks. At the end of this procedure, we obtain a "topologically and electrically" characterized power grid, which can be used for simulating cascading failures within a DC power flow approximation of the power network behavior.

In what follows, the symbol $G$ will be used to denote the complete power transmission network configuration, i.e., its network topology, the loads, the generators, and the link impedances and capacities.

### 3. Cascading Failure Model

The procedure adopted for simulating cascading failures on a random network $G^0 = G$ generated by the algorithm described above bears some similarities with that employed in Ref. [12] (see Algorithm 1). An initial failure in line $l \in E$, potentially triggering the cascade, is chosen. At step 1 of Algorithm 1, the algorithm computes the new power flows in the transmission network $G^1 = (V, E - \{l\})$, i.e., the initial network without line $l$. Then, each iteration of the **FOR** loop in Algorithm 1 corresponds to a line disconnection event in the cascading failure simulation, which occurs whenever a line becomes overloaded. At step 2, the algorithm re-dispatches the loads and the generation in order to keep all the possible islands in the network balanced; in fact, the islands form when the line failures break the original network $G$ in multiple connected components, which might have an excess in power supply or demand. Recall that in order to compute the DC power flow, each connected component needs to be balanced in terms of power supply and demand. The re-dispatch algorithm adopted in this work is described in Appendix A. At step 3, the new power flows are computed and, at step 4, a line outage model (described in Appendix A) accounting for the dynamic evolution of the power flows within the network lines is adopted to identify the next disconnected lines (if any). At step 5, the network configuration is updated. The procedure is iterated until lines become overloaded; otherwise, the cascade simulation stops.

---

**Algorithm 1. Procedure adopted for simulating cascading failures on a random network.**

---

**INPUTS:** Power network $G^0 = G$ and initial line failure $l \in E$

1.    $G^1 = (V, E - \{l\})$, compute vector $\boldsymbol{F}^1$ of power flows in $G^1$

**FOR** $r = 1, 2 \ldots$ **DO**

2.    Adjust load and generation (re-dispatch)

3.    Compute $\boldsymbol{F}^r$ power flows vector in $G^r$

4.    Set $O^r$, i.e., the set of lines that become outaged at round $r$

**IF** $|O^r| \geq 1$

5.    Set $G^{r+1} = (V, E^r - O^r)$

**OTHERWISE**: END

---

The DC power flow, re-dispatch strategy and outage model are described in Appendix A.

## 4. Results

### 4.1. Case Study Description

We exploit the methodology and algorithms described in the previous Sections for performing statistical analysis of the behavior of power transmission networks with respect to cascading failures. In order to do so, we resorted to the proposed algorithm for generating $N_s = 5000$ and $N_s = 800$ power grids statistically similar to the IEEE-14 and IEEE-118, respectively (from the Power System Test Case Archive, UWEE). Both the number of nodes and the power supplied by each generator are the same as those of the IEEE reference grid. The properties of the IEEE-14 and IEEE-118 networks taken for reference are summarized in Table 1.

**Table 1.** Main properties of the IEEE-14 and IEEE-118 test cases.

|  | **IEEE-14** | **IEEE-118** |
|---|---|---|
| Number of buses | 14 | 118 |
| Number of links | 21 | 184 |
| Number of generators | 2 | 19 |
| Number of transmission nodes | 0 | 35 |
| Number of loads | 12 | 64 |
| Total power generated | 272.4 MW | 4377.4 MW |

We set $\rho = 10^{-1}$, $\varepsilon = 10^{-1}$ so as to study the behavior of grids operating close to their limit, in in an attempt to represent the conditions of old-generation power grids overburdened by the continuously increasing power demand [3]. On the other hand, we set $\alpha = 10^{-5}$ within a trial-and-error procedure aiming to minimize the occurrence of simultaneous overloads during the cascades. This procedure is then applied to all the links in the sampled network configuration. Due to the deterministic nature of the cascade propagation, each simulation is completely identified by the pair $(G_j, i)$, where $G_j$ $(j = 1, \ldots, N_s)$ is the sampled initial network configuration and $i$ is the index of the initial link removed from $G_j$. By so doing, for each $G_j$ we obtain $m_j$ cascading failure simulations, yielding a total of 98,213 cascading failures in the $N_s = 5000$ IEEE14-like network realizations and 139,245 in the $N_s = 800$ IEEE118-like network realizations.

In order to be able to quantify the intensity of a cascading failure event, we introduce the following indicator, computed at the end of each cascading failure simulation $(G_j, i)$:

$$FO_{G_j,i} = \frac{N^{out}_{G_j,i}}{m_j} \tag{4}$$

i.e., the fraction of lines cut due to overloads, where $N_{G_j,i}^{out}$ is the number of outages due to overloads at the end of a cascading failure event initiated by the removal of link $i$ in the network $G_j$. Another indicator of the cascading failure intensity is defined as follows [18]:

$$FLS_{G_j,i} = \frac{\left(PD_j^{total}(0) - PD_{G_j,i}^{total}(r_{last})\right)}{PD_j^{total}(0)} \quad (5)$$

i.e., the final fraction of load shedding, where $PD_{G_j}^{total}(0)$ is the total power delivered in the network $G_j$ before the initial link $i$ removal, and $PD_{G_j,i}^{total}(r_{last})$ is the power still successfully delivered at the last iteration of Algorithm 1, i.e., at the end of the cascade of failures initiated by the removal of line $i$.

Finally, while $FO_{G_j,i}$ and $FLS_{G_j,i}$ are important measures for quantifying the severity of a cascading failure scenario, the cascading failure duration $T_{G_j,i} = t^{r_{last}} - t^0$—in terms of the number of time steps elapsed from the time of the initial line $i$ failure, $t^0$, to the end of the cascading event, $t^{r_{last}}$—plays, in general, an important role for cascade control and mitigation. In all the scenarios simulated, with the purpose of facilitating the display of the results in logarithmic scales and with no loss of generality, the initial time step at which the initiating line failure event occurs is arbitrarily set to $t^0 = 10$.

### 4.2. Statistical Analysis

In what follows, we analyze the statistical properties of the randomly generated network configurations from the point of view of the intensities of the simulated cascading failure scenarios. The aim is that of capturing the main recurrent behaviors and, possibly, finding relationships with some intrinsic properties of the generated networks.

Figure 1 shows the histograms of the $FO_{G_j,i}$ of the cascading failure scenarios obtained for the IEEE-14-like (a) and the IEEE-118-like (b) networks, respectively. It can be noted that despite the large difference in the number of nodes of the two network typologies, there seem to exist a few common features. First, the range of variability of $FO_{G_j,i}$ for both types of networks is very similar, i.e., approximately (0, 60%) and (0, 70%) for IEEE-14-like and IEEE-118-like networks, respectively. This suggests the existence of a physical limitation in the propagation of the cascading failures, preventing larger disconnections of the networks. A possible motivation probably lies in the fact that when the loss of lines due to overloads and unbalanced islanding becomes very large with respect to the network size, the remaining amount of load that can be satisfied becomes so small that the power flowing in the remainder of the network is not enough to trigger further outages due to overload. A similar saturation effect was identified also in Ref. [32] in a similar context, where a truncated branching process was employed to model the number of line overloads during a failure cascade. Then, $FO_{G_j,i}$ follows a bimodal distribution, with a separation between the occurrences of cascading failures affecting small portions of the network and those leading to larger consequences in terms of lines disconnections. This behavior is less evident for IEEE14-like networks due to the fact that the average number of lines of the $N_s = 5000$ IEEE14-like network configurations (19.6) is rather low, so the two modes of the distribution tend to overlap. On the other hand, in the case of the IEEE118-like networks, the average number of lines is 173.7 and the two modes are well separated by a range of $FO_{G_j,i}$ values, the correspondence of which features almost no cascading scenarios.

The motivation of the bi-modal behavior of the scenario distributions probably lies in the superposition of two competing processes. The first process is the natural extinction of the cascading overloads due to the network reaching a new sustainable equilibrium point. Figure 2 shows the histograms of the $FO_{G_j,i}$ in scenarios which do not lead to any islanding (IEEE14-like (a) and IEEE118-like networks (b)); in both network typologies, the cascading failures extinguish for $FO_{G_j,i} < 10\%$. The second process is the formation of isolated islands due to the separation of portions of the transmission network including more buses and lines; this process, in turn, boosts the further propagation of multiple

parallel cascades in the separated islands (provided the islands have at least one generator; otherwise, they undergo a full blackout). This is due to the simple re-dispatch strategy adopted in this work, which does not take into account the capacities of the lines; in fact, in the event that multiple large islands form during the cascading failure, in an attempt to balance the individual island power generation and demand, this can easily give rise to new overloads, thus triggering the further propagation of the cascading failures within the islands. This process gives rise to the second mode of the histogram, shifted towards a larger number of overloading events, as shown by Figure 2, which reports the histograms of the $FO_{G_j,i}$ associated with scenarios leading to islanding of any size (IEEE14-like (c) and IEEE118-like networks (d)). Indeed, the process can start only when the number of lines disconnected due to overload becomes large enough, with respect to the network size, to isolate significant portions of the network. Figure 3 shows the histograms of $FO_{G_j,i}^{isl}$, i.e., the fractional overload, within whose correspondence the network experiences the formation of the largest power island of the entire cascade event. Note that by the largest island in the grid, we mean either the largest separate aggregation of nodes and links if it does not contain generators or the second largest island containing generators, since we formally consider the first as a direct evolution of the original grid. In the IEEE118-like networks, the formation of the larger islands starts at approximately $FO_{G_j,i}^{isl} = 5\%$ and finishes between $FO_{G_j,i}^{isl} = 40\%$ and $FO_{G_j,i}^{isl} = 50\%$ (Figure 3b), giving rise to the delayed second mode of the histogram of Figure 1 due to the delayed cascading failure, as explained above. A similar behavior can also be observed for IEEE14-like networks (Figure 3a), although its interpretation from the histograms is more difficult due to the small number of lines involved.

However, a small contribution to the first mode of the histogram in Figure 3b still exists in the case of the IEEE118-like networks, due to cascades involving some islanding. In order to further investigate this behavior, Figure 4b shows a scatterplot of $FO_{G_j,i}$ versus the size, in terms of number of nodes, of the largest island (see the definition given above) formed during the corresponding cascading failure; the grey intensity of the points in the scatterplot is the $\log_{10}$ of the number of observed occurrences. It can be seen that the first mode of the histogram receives contributions of scenarios involving the formation of islands whose larger size is generally smaller than 10, whereas the second mode of the histogram is due to occurrences of scenarios leading to larger islanding, with no apparent correlations between the size of the larger island formed and $FO_{G_j,i}$. A conclusive topological motivation of this small "early" contribution to islanding is still to be found; on the other hand, the occurrences of scenarios of this kind are almost negligible with respect to those associated with the two main behaviors discussed earlier.

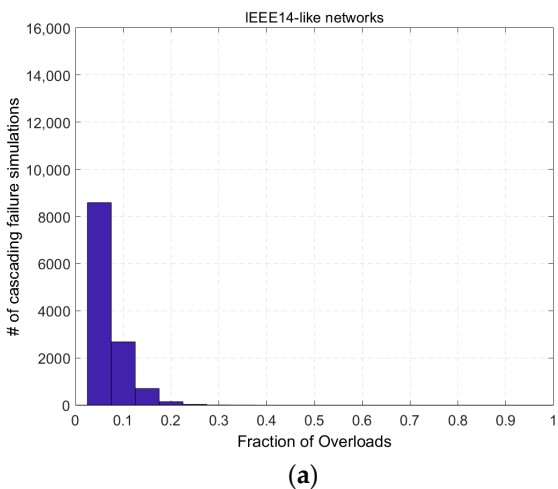

(**a**)

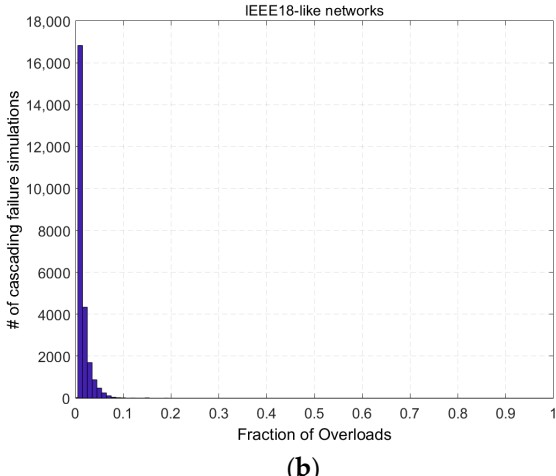

(**b**)

**Figure 2.** *Cont.*

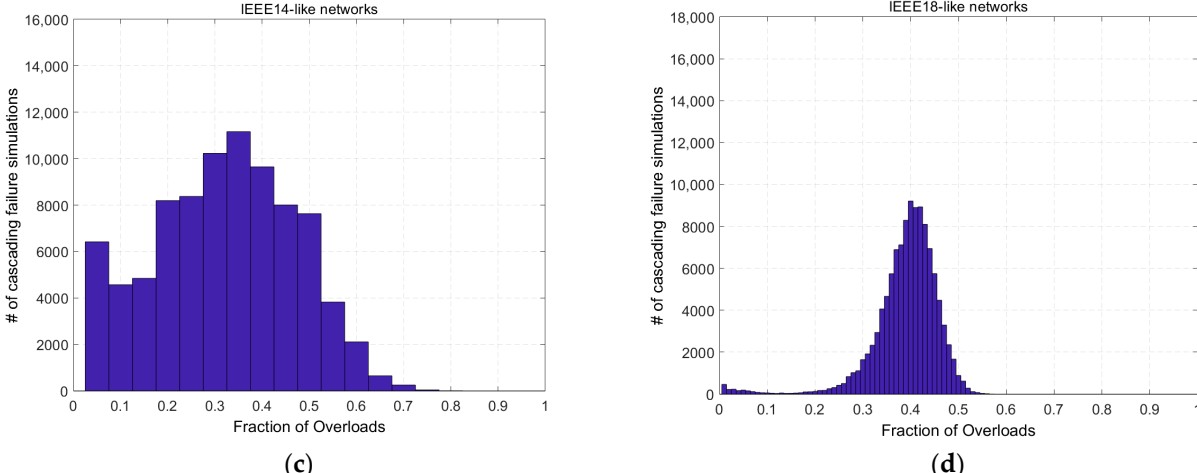

**Figure 2.** (**a**–**d**): Histograms of $FO_{G_j,i}$ of cascading failure scenarios with no islanding and with islanding for the IEEE14-like (**a**,**c**) and the IEEE118-like (**b**,**d**) random networks.

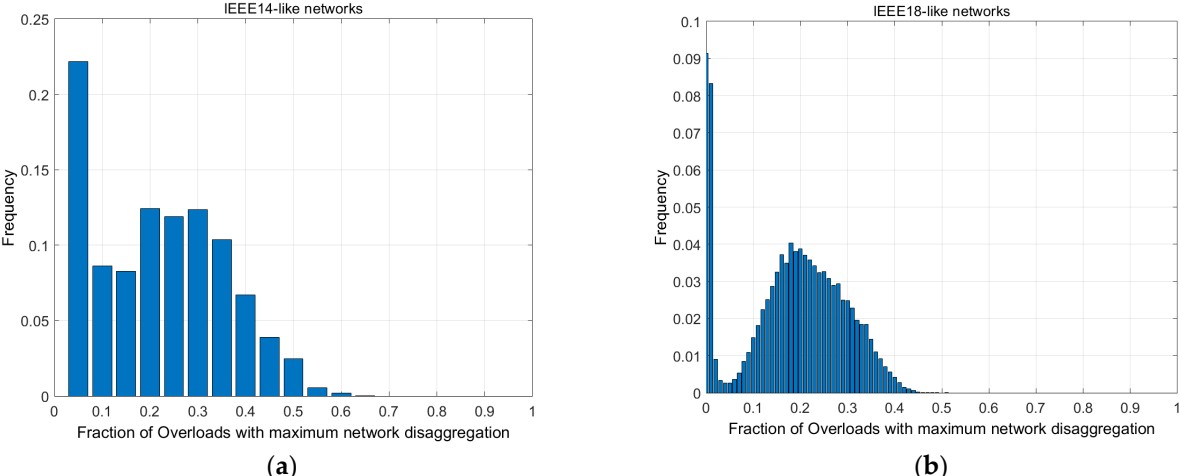

**Figure 3.** (**a**,**b**): Histograms of $FO_{G_j,i}$, the correspondence of which features the formation of the largest island of the entire cascade event for the IEEE14-like (**a**) and IEEE118-like (**b**) random networks.

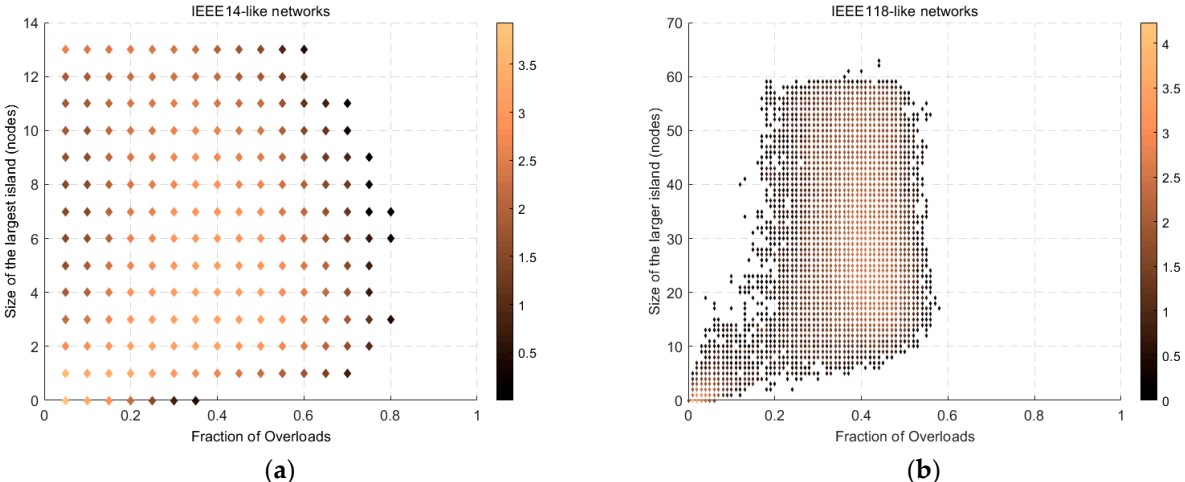

**Figure 4.** (**a**,**b**): Fraction of overloads versus size, in terms of number of nodes, of the largest island for the IEEE14-like (**a**) and the IEEE118-like (**b**) random networks. The grey scale is proportional to the logarithm of the number of simulations.

For the IEEE14-like networks, the same scatterplot (Figure 4a) does not allow us to draw similar conclusions, probably due to the same problem of overlapping behaviors illustrated above.

With reference to the second indicator introduced at the beginning of this Section, Figure 5 shows the histograms of the $FLS_{G_j,i}$ obtained for the IEEE14-like (a) and the IEEE118-like networks (b). As already shown in Figure 1a,b, in both types of networks, cascading failures rarely lead to fractional overloads $FO_{G_j,i}$ larger than 60–70%, but in smaller systems, the same portion of disconnected lines leads to a broader range of fractional power losses, $FLS_{G_j,i}$, as shown in Figure 6a. In fact, a power network with only 14 buses is so small that in some of the configurations generated, even the failure of a single line (without any further propagation) can potentially jeopardize the entire power distribution. The bi-modal behavior observed for the distribution of the $FO_{G_j,i}$ at the end of the cascading events (Figure 1b) is thus lost when considering the distribution of the corresponding $FLS_{G_j,i}$ for the IEEE14-like networks because the correlation between $FO_{G_j,i}$ and $FLS_{G_j,i}$ is weak, as shown in the scatterplot of Figure 5c. On the other hand, for the IEEE118-like networks (Figure 5b), the bi-modal behavior is shown by both $FO_{G_j,i}$ and $FLS_{G_j,i}$ due to the larger correlation between $FO_{G_j,i}$ and $FLS_{G_j,i}$ (scatterplot of Figure 5d).

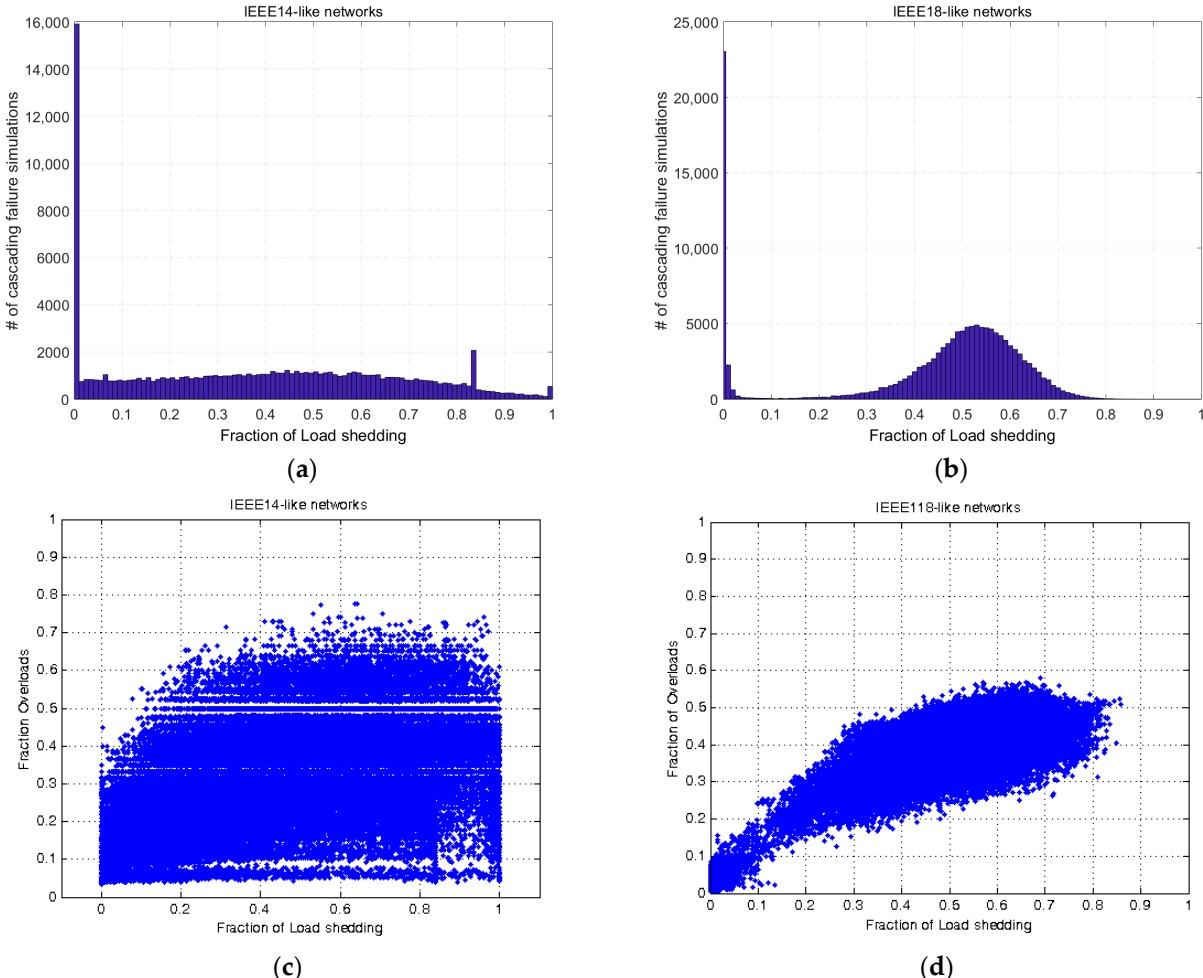

**Figure 5.** Histograms of $FLS_{G_j,i}$ of cascading failure scenarios for the IEEE14-like (**a**) and the IEEE118-like (**b**) random networks. Corresponding scatterplots of the simulations in the ($FLS_{G_j,i} - FO_{G_j,i}$) plane for the IEEE14-like (**c**) and the IEEE118-like (**d**) random networks.

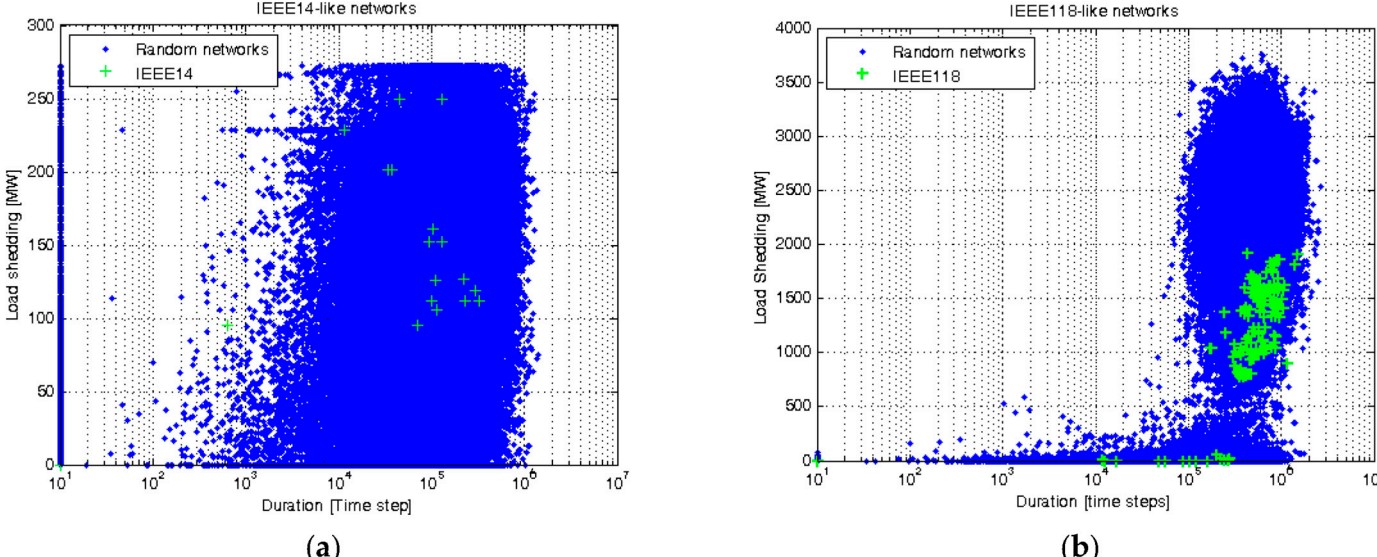

**Figure 6.** Scatterplots of the simulations in the $(\log_{10} T_{G_j,i} - FLS_{G_j,i})$ plane for the IEEE14-like (**a**) and the IEEE118-like (**b**) random networks. The green crosses indicate simulations performed on the original IEEE14 (**a**) and IEEE118 (**b**) reference networks.

Figure 6 shows the scatterplots of the cascading failure scenarios in the plane $(\log_{10} T_{G_j,i} - FLS_{G_j,i})$ for the IEEE14-like (a) and the IEEE118-like (b) networks. These plots are important for quickly visualizing cascading scenarios that could represent a challenge for an automatic control strategy aiming to mitigate the propagating events, i.e., those characterized by short durations, $T_{G_j,i}$, and large fractional load shedding, $FLS_{G_j,i}$. The points located at $\log_{10} T_{G_j,i} = 1$ (Figure 6a) correspond to scenarios wherein the initial failure does not propagate but gives rise to a large power loss. The duration of these scenarios is $T_{G_j,i} = 10$, which corresponds to the arbitrarily chosen time of occurrence of the first failure; in fact, the model (13) for the power grid inertia obviously intervenes after the first failure occurrence. As verified by the authors but not shown here for brevity's sake, these scenarios occur in a few "pathological" networks randomly containing evident structural vulnerabilities (for example, both generators connected to the rest of the network by only one line) and unrealistic topologies (for example, tree-like topologies). Thus, in these cases, no effective mitigation strategies could be devised without a preventive topological re-design of the network.

The simulations of cascading failures in IEEE118-like networks (Figure 6b) appear to be more concentrated than those obtained in IEEE14-like networks (Figure 6a). This is due to the fact that the sample space of the IEEE118-like networks is by far larger than that of the IEEE14-like networks, since the space dimension increases exponentially with the number of nodes of the grid, so that the sample size ($N_s = 800$ random topologies) is relatively small and no critical configurations are included. On the other hand, the smaller sample space and, at the same time, the larger sample size ($N_s = 5000$) allow for better exploration of the IEEE14-like networks. Larger sizes of the IEEE118-like samples would require computational efforts beyond the scope of this work.

The scatterplots of Figure 6 show also the cascading failure scenarios obtained with the original IEEE14 and IEEE118 networks (green crosses). It can be seen that even with the rather crude re-dispatch strategy adopted in this work, all the points corresponding to the simulated scenarios lie well within the main "clouds", without showing any criticalities. This confirms the expectation, since the original IEEE networks of reference were actually extracted from real portions of the US power transmission network (from the UWEE Power Systems Test Case Archive); as such, they must have been designed to be robust with respect to this kind of failure.

As a final remark, note that the re-dispatch policy adopted is purposely simplistic to allow for the generation of a large number of network configurations in reasonable computational times; this is in order to explore a broad region of the cascading failure scenario space. On the other hand, the adoption of such a re-dispatch strategy with a minimal number of actions involved allows for the effects of the network topology and electrical characteristics on the cascading failure propagation to be enhanced. This is useful for identifying the weaknesses of an individual network or of a class of networks with respect to these characteristics. Finally, the conservative nature of the re-dispatch logic employed in this work ensures that the protection/control system, optimized based on the outcomes of these analyses, will be highly robust against cascading failures.

### 4.3. Cascading Failure Control and Mitigation Effort Measures

We introduce the following measure associated with the individual cascading event $i$ to rank the scenarios in the scatterplots of Figures 5 and 6 in terms of the effort they require to be controlled and mitigated with respect to cascading failures:

$$S_i = FLS_{G_j,i} * \frac{\log_{10} T_{max}}{\log_{10} T_{G_j,i}} \tag{6}$$

where $T_{max} = 10^8$ time steps is a bound chosen so as to be larger than all the cascading failure durations simulated. Taking the logarithm of the cascade durations is a convenient choice based on the observation that the metric would readily allow us to identify critical areas in scatterplots of the kinds of Figures 5 and 6, where the adoption of a logarithmic scale for $T_{G_j,i}$ is needed to capture the general behavior of the networks. On the other hand, a ranking of the control and mitigation effort based on this metric would be the same without the use of the logarithm. The proposed metric accounts for both the cascade propagation time $T_{G_j,i}$ and the fraction of load loss $FLS_{G_j,i}$, which are two major objectives of control and mitigation. More precisely, the metric measures an average "velocity" of load loss in the development of the cascading event: the larger this velocity is, the more effective a control strategy must be in order to limit the spreading of the cascade and terminate the propagation. Note that for both IEEE14-like and IEEE118-like networks, $S_i$ can assume values in the range $(0, 8)$, with larger control and mitigation efforts required in association with with higher values.

By averaging the values of the metric $S_i$ over all the scenarios pertaining to the individual sampled configuration $G_j$, for each $j = 1, \ldots, N_S$, it is possible to obtain a new metric for the control and mitigation efforts of a given network with respect to cascading failures due to a single initiating event. Specifically, we compute the following:

$$MS_j = \frac{1}{Nl_j} \sum_{i=1}^{Nl_j} S_i \tag{7}$$

Furthermore, the dispersion of $S_i$ around its mean for each configuration $G_j$ is as follows:

$$\sigma S_j = \sqrt{\frac{1}{Nl_j - 1} \sum_{i=1}^{Nl_j} \left(S_i - MS_j\right)^2} \tag{8}$$

This represents a measure of the robustness of the design configuration, since a strong average score $MS_j$ could also be achieved when a few very critical scenarios characterized by large values $S_i$ occurred, thus rendering the network vulnerable. Taken together, the pair $\left(MS_j, \sigma S_j\right)$ provides a more accurate representation of the overall behavior of configuration $G_j$ with respect to the propagation of cascading failures.

Figure 7 shows the sorted values of $MS_j$ (blue crosses) and the corresponding $\sigma S_j$ (red circles), respectively, obtained for the $N_S = 5000$ IEEE14-like (a) and the $N_S = 800$ IEEE118-like (b) networks. The ranges of variability of these two metrics are larger for IEEE14-like than for the IEEE118-like configurations. This is due to the fact that (i) as said

before, the sample size of the IEEE14-like grids is relatively much larger than that of the IEEE118-like ones, so that many more configurations at the extremes can be investigated. (ii) Smaller power grids are made of fewer components, so that they tend to be more sensitive to the failure of one of them.

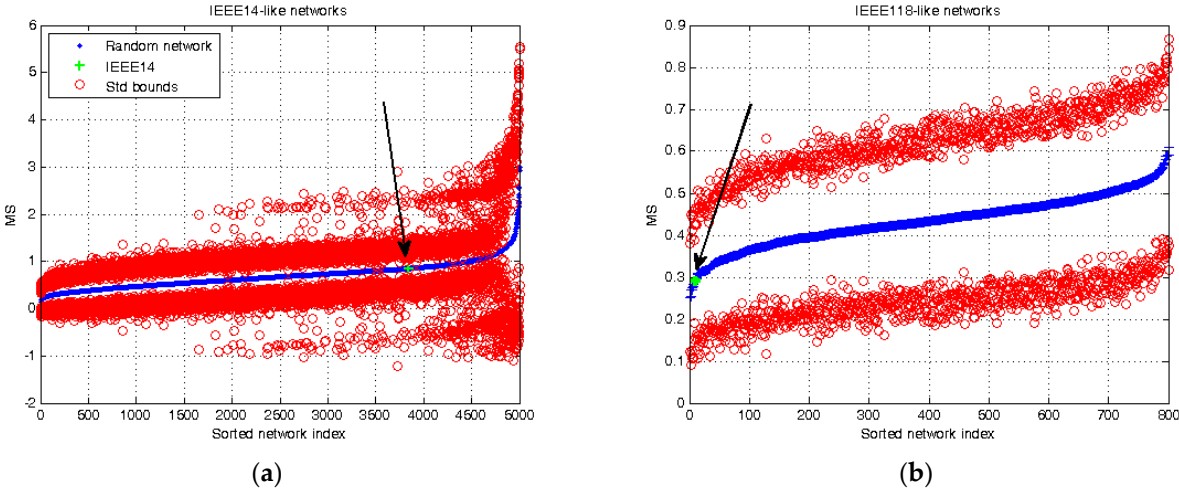

**Figure 7.** Mean metric $MS_j$ (blue crosses) and its standard deviation $\sigma S_j$ bands (red circles) for the IEEE14-like (**a**) and the IEEE118-like (**b**) random networks. The green crosses mark the positions of the original IEEE networks.

The plots of Figure 7 also show the values obtained for the reference IEEE14 and IEEE118 power transmission networks (green crosses). Interestingly, the original IEEE14 and IEEE118 networks are ranked 3846 out of $N_S + 1 = 5001$ and 10 out of $N_S + 1 = 801$ random networks, respectively. Thus, apparently, the reference network IEEE118 is more robust with respect to cascading failures than the reference IEEE14, although this result may be due to the same issue of sample space dimension and coverage by the simulations mentioned above.

Surprisingly, the standard deviations $\sigma S_j$ of the metric values in Figure 7a for the IEEE14-like networks can be roughly clustered into two families, as opposed to the case of the IEEE118-like grids. This behavior is likely due to the presence or the absence of at least one connection whose initial failure causes the loss of at least 50% of the whole demanded power in the network; the cascading failure sequences originating from the failures of these connections are made out of the initial failure event alone and thus have a very short time duration (fixed at 10, as shown before), such that the associated metric $S_i$ tends to assume values slightly above the average. These sequences are then characterized by large values of $\sigma S_j$. The clustered behaviors of the metric $\sigma S_j$ are further highlighted in the scatterplot $(MS_j - \sigma S_j)$ of Figure 8a, where the points corresponding to the sequences in the IEEE14-like networks with more than 50% of load shedding due to the initial failure (green squares) are shown to correspond to the points in the cluster characterized by the large $\sigma S_j$. No such behaviors can be observed for the IEEE118-like networks (Figure 8b), where their corresponding $\sigma S_j$ values have a general behavior similar to the that of the first cluster of low values for the IEEE14-like networks. Again, this is due to the poor exploration of the sampling space achievable for systems of increasing size. It is likely that if we were able to generate a few orders of magnitudes more sample of IEEE118-like networks, it would be possible to observe the same anomalous configurations with strong structural deficiencies. In conclusion, the metric $\sigma S_j$ could in principle also be used for identifying power grid configurations with obvious structural vulnerabilities among those automatically generated, such as, for example, those not meeting the $N - 1$ reliability criteria [22].

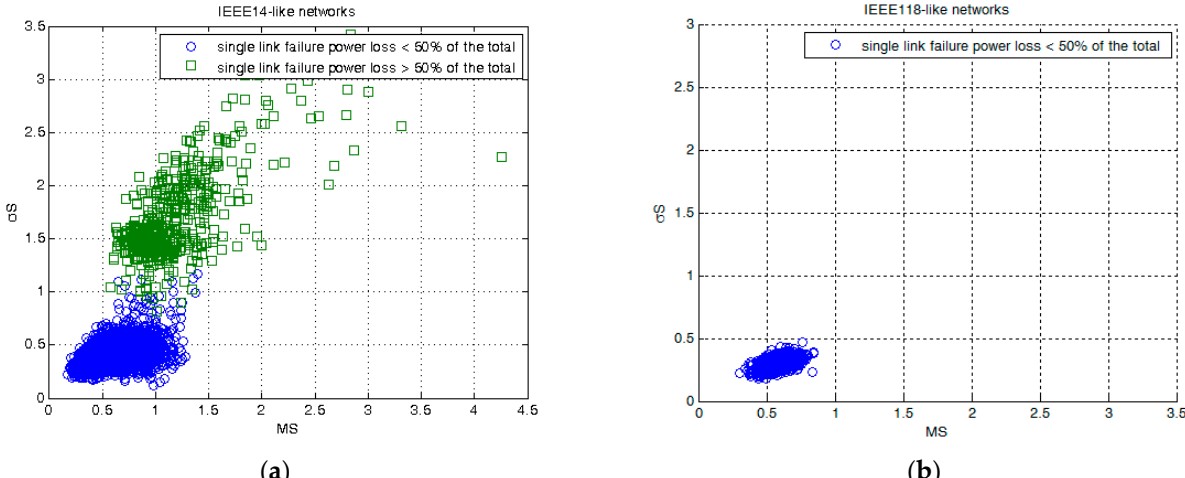

**Figure 8.** Scatterplots of the simulations in the ($MS_j - \sigma S_j$) plane for the IEEE14-like (**a**) and the IEEE118-like (**b**) random networks.

### 4.4. Sensitivity Analysis and Genetic Algorithm Optimizations

Besides the cases of poorly designed networks with obvious flaws, other configurations with no evident problems can be characterized by rather large values of $MS_j$. This is interesting because it shows that even more subtle differences in configuration can have a strong influence on the overall behavior of the system. Indeed, the identification of which kinds of differences lead to such a degradation of the network robustness with respect to cascading failures is very important, but it is not an easy task, due to the large number of features involved in the network sampling process, i.e., the topology, the loads, the generation distribution and the impedances of each line. More specifically, the difficulties are related to (i) their distributed nature, especially with regard to topologies; (ii) their high interdependences (recall that in the proposed sampling algorithm, the loads/generators and the impedances depend on the sampled topology); and (iii) the potentially associated computational expenses.

We propose to exploit the control and mitigation efforts metric definitions given above and the sampling framework developed used in this work to perform a crude sensitivity analysis aiming to identify which features most influence the behaviors of the networks with respect to cascading failures.

For computational issues, but with no loss of generality, we make reference only to the IEEE14 network. First, we sample 5000 new random topologies and the corresponding link impedances, but we keep the power demands and generations of each of the 14 nodes of the sampled networks fixed to those of the reference IEEE14 grid. Ideally, in order to isolate the single contributions to the control and mitigation effort metrics, the topologies and the impedances should be varied one at a time, but this is impossible for the topologies alone since a change in the topology would inevitably also change the impedance distribution. Then, we rank the synthetic networks according to the values of the metric $MS_j$, thus obtaining the plot in Figure 9 (blue crosses), where the standard deviations $\sigma S_j$ are also shown (red circles); note that the procedure is similar to that which led to Figure 7, but the values of $MS_j$ now depend only on the topologies and the impedances. A visual comparison with the results in Figure 7a, obtained by sampling all the features together, does not highlight any particular difference, showing that the majority of the variability of the metric $MS_j$ is due to the topology and impedance distributions. This confirms the aforementioned observations of single topologies with no apparent deficiencies leading to catastrophic cascades with very large values of $MS_j$.

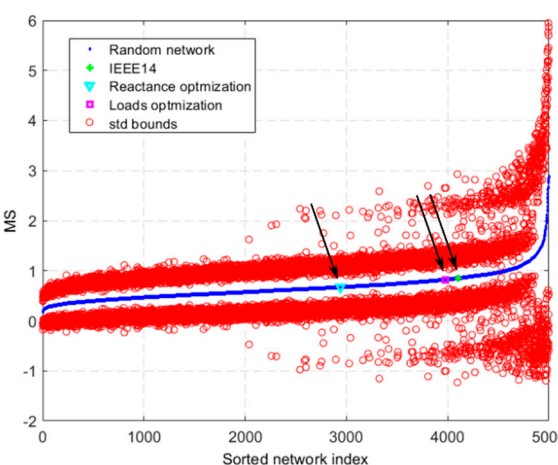

**Figure 9.** Mean metric $MS_j$ (blue crosses) and its standard deviation $\sigma S_j$ bands (red circles) for the IEEE14-like networks and the optimized grids (impedances: light blue triangle; loads: purple square). The green crosses mark the positions of the original IEEE networks.

The intuitive sensitivity analysis can now be further carried on by investigating the effects of the impedance and load distributions alone. For the reasons explained before, unfortunately, the effects of the impedances can be observed only with a fixed topology; thus, we perform the analysis on the original IEEE14 configuration. However, instead of obtaining curves similar to those of Figure 7a,b, which would have a rather limited meaning, since the topology is fixed, we propose a more operative and useful sensitivity approach based on the minimization of the value of $MS_{IEEE14}$ by separately optimizing the line impedances' allocation and the load distribution on the nodes.

Even though the network is small, the dimensions of the minimization search spaces (21 for the impedances and 14 for the loads) are such that classical gradient descent-based optimization algorithms could be inefficient, and the problem surely becomes more and more burdensome as the size of the network increases. Thus, here, we resort to a genetic algorithm [33].

In order for the values of the system parameters to remain physically meaningful, we allow a 10% maximum variation around their values in the original IEEE14 network during the optimization. The results of the two GA optimizations are shown in Figure 9 by light blue triangles (impedance optimization) and purple squares (load configuration optimization). In both cases, the GA optimizations lead to an improvement of the original IEEE14, suggesting that the control and mitigation requirements of a given network with respect to cascading failures can be improved by re-designing both the load distribution and the impedance scheme. Note that from a sensitivity analysis point of view, a significantly larger improvement can be achieved by varying the impedance distribution alone. The improvements achieved are not very large, but this was to be expected due to the fact that (i) the IEEE14 reference network represents a portion of an actual power transmission network, that has been optimally designed to avoid failure propagations and (ii) the search ranges for both impedances and loads are rather narrow. As already noted above, Figure 9 further shows that the maximum improvements on the single IEEE14 configuration due to the impedances and loads optimizations are smaller than the potential ones achievable by modifying the topology of the network.

Although the impedances and load optimizations could not be performed for the IEEE118 original network due to the computational constraints mentioned above, its margins of improvement with respect to the metric $MS_{IEEE118}$ are apparently much smaller than for the IEEE14 grid, as shown in Figure 7b. However, before a final conclusion can be drawn, a broader exploration of the sampling space is required in order to possibly increase the range of variability of $MS_{IEEE118}$.

## 5. Conclusions

In this work, we have investigated some statistical properties of power transmission networks with the general objective of identifying common strengths and weaknesses with respect to their cascading failure behavior. In order to do so, we have integrated a random power grid generator with a cascading failure simulator based on a DC approximation of the power flows. The integrated algorithm has allowed us to perform a systematic analysis of the cascading failure dynamics within a broad set of networks that are individually different in terms of power grid topologies, loads configurations, generators configurations and line impedances but bear the same statistical properties typical of actual power transmission grids. In particular, in this work, we have referred to both IEEE14- and IEEE118-like networks.

The analysis has led to the identification of an unexpected bi-modal behavior of the cascading failures, leading to load shedding, with respect to the final number of lines disconnected due to overloading. It has been shown that this bi-modality is strictly correlated to the formation of large islands during the cascade propagation, which, under the re-dispatch strategy adopted in this work, amplifies the network damage in terms of load losses.

Then, we developed a new metric for quantifying the control and mitigation requirements of individual scenarios with respect to cascading failures; this metric accounts for the duration of a cascading failure event initiated by a single-line failure and its consequences in terms of final load shedding. By averaging the values of the metric over all possible line failures in a single-grid model, we obtained a new metric for the robustness of a whole network with respect to cascading failures; the standard deviation of the metric can also be interpreted as a metric itself, since it captures the presence of weak lines whose failures lead to anomalous propagating behaviors in power networks, which otherwise appear to be robust. The metric has been shown to be able to correctly identify as critical those randomly generated scenarios and/or configurations that present evident design flaws; however, at the same time, it also identifies those scenarios/configurations with more subtle and unexpected deficiencies, which are otherwise very difficult to capture. The calculation of the proposed metric has also allowed us to perform a rough sensitivity analysis of the features most influencing the cascading failure behavior of power transmission networks, which confirmed the expected importance of the topology and of the correlated impedances' distribution.

Finally, we have shown that the proposed metrics can be effectively exploited within a genetic algorithm search scheme for the identification of optimal improvements to an existing power grid in terms of both line impedances and loads at the nodes. The new metrics can then be effectively included in practical optimizations, where the necessary modifications to national and over-national power grids must be chosen by taking into account several other possibly conflicting objectives such as economics, congestion issues, political considerations, etc.

There are many other potential uses of the computational approach developed in this work. One of the most promising seems to be that of automatically generating and selecting critical scenarios/configurations to be used as worst-case scenarios for testing, validating, and improving the robustness of the control or mitigation strategies to be adopted during cascading failure events; control and mitigation effort metrics, such as those introduced in this work, may serve to compare the performances of different controllers, possibly also accounting for associated economic requirements.

Current and future research efforts are focused on strengthening the applicability of the proposed approach, which is mainly methodological, to practical problems. This entails, first of all, modifying the random network sampling algorithm in order to be able to generate only $N-1$ secure configurations and, secondly, accessing more powerful and parallelized computational resources in order to be able to study the dynamics of more realistic large power transmission networks.

**Author Contributions:** Conceptualization, F.C. and E.Z.; methodology, F.C.; software, F.C.; validation, F.C. and L.L.; formal analysis, F.C.; investigation, F.C. and L.L.; resources, E.Z.; data curation, F.C. and L.L.; writing—original draft preparation, F.C. and L.L.; writing—review and editing, F.C. and L.L.; visualization, F.C. and L.L.; supervision, E.Z. All authors have read and agreed to the published version of the manuscript.

**Funding:** This research received no external funding.

**Data Availability Statement:** Dataset available on request from the authors.

**Conflicts of Interest:** The authors declare no conflict of interest.

## Appendix A

This appendix describes in detail the steps in Template 1.

### *Appendix A.1. DC Power Flow*

The vector $f^r$ at each iteration of Algorithm 1 is computed with the DC power flow MATLAB function MATPOWER [31]. The DC power flow model is a linear approximation of the AC power flow model, which relies on the following assumptions [22]:

- Flat voltage profile: all bus voltage phasors are 1.0 per unit in magnitude.
- The lines are lossless or, equivalently, line resistance is neglected:

$$\forall \, l \in E \; z_l = res_l + ix_l \approx ix_l$$

- Voltage angles differences are small enough that

$$\forall \, i,j \in V \; \sin\left(\theta_i - \theta_j\right) \approx \theta_i - \theta_j$$

Under the above assumptions, let $\theta_i$ be the voltage angle at bus $i$ and $\boldsymbol{\theta}$ the $N$ dimensional vector of voltage angles, where a component is chosen to be equal to $0$ (reference bus). Let $b_{ij}$ be the susceptance of the line joining bus $i$ and bus $j$. The $N \times N$ matrix at the $r$-th iteration of Algorithm 1 $\boldsymbol{B}^r$ is defined as follows:

$$B_{ii}^r = \sum_{bus \; j \; connected \; to \; bus \; i} b_{ij}^r$$

$$B_{ij}^r = -b_{ij}$$

The DC load flow equations are as follows:

$$\boldsymbol{P}^r = \boldsymbol{B}^r \boldsymbol{\theta}^r \tag{A1}$$

where $\boldsymbol{P}^r$ is the vector of the node power outputs, which sums to zero due to the balance constraint (A1). The matrix $\boldsymbol{B}^r$ has rank $N - 1$: thus, removing the row corresponding to the reference bus and taking the inverse, we obtain

$$\boldsymbol{\theta}^r = \boldsymbol{X}^r \boldsymbol{P}^r \tag{A2}$$

Finally, the power flow in line $l$ connecting bus $i$ and $j$ is found:

$$f_l^r = b_{ij}^r \left( \theta_i^r - \theta_j^r \right) \tag{A3}$$

For further details on the DC power flow approximation, the interested reader may refer to Ref. [22].

### *Appendix A.2. Re-Dispatch Strategy*

During the propagation of the cascading failures, the transmission grid could become separated in multiple connected components, also called islands. The newly formed

islands, in principle, do not meet the power balance Equation (A1), thus not allowing further DC computation. In order to overcome this issue a simple power re-dispatch routine is embedded in the simulator. Considering the island $I \subset G$, where $(Gen_I, Load_I)$ are the sets of generator and load buses in $I$, at each iteration $r$, three different unbalanced possibilities could arise:

1. The power supply does not meet the demand but the generators have enough reserve to accommodate the surplus:

$$\sum_{i \in Gen_I} PG_i^{r-1} < \sum_{i \in Load_I} PD_i^0 \ and \ \sum_{i \in Load_I} PD_i^0 < \sum_{i \in Gen_I} PMAX_i$$

Then, the power production of each generator belonging to $I$ is increased proportionally:

$$\forall i \in Gen_I \ : \ PG_i^r = PG_i^{r-1} + \frac{PMAX_i - PG_i^{r-1}}{\sum_{i \in Gen_I} PMAX_i - \sum_{i \in Gen_I} PG_i^{r-1}} (\sum_{Load_I} PD_i^0 - \sum_{Load_I} PG_i^{r-1}) \tag{A4}$$

$$\forall j \in Load_I \ : \ PD_j^r = PD_j^0 \tag{A5}$$

2. The power supply does not meet the demand and does not have enough reserve to meet the surplus:

$$\sum_{i \in Gen_I} PG_i^{r-1} < \sum_{i \in Load_I} PD_i^0 \ and \ \sum_{i \in Load_I} PD_i^0 > \sum_{i \in Gen_I} PMAX_i$$

Then, the power demand of each load belonging to $I$ is decreased proportionally and the power supply is fixed to the maximum limit:

$$\forall i \in Load_I : \ PD_i^r = PD_i^0 \frac{\sum_{i \in Gen_I} PMAX_i}{\sum_{i \in Load_I} PD_i^0} \tag{A6}$$

$$\forall j \in Gen_I : \ PG_j^r = PMAX_j \tag{A7}$$

3. Meanwhile, if in the island $I$ the power supply exceeds the demand,

$$\sum_{i \in Gen_I} PG_i^{r-1} > \sum_{i \in Load_I} PD_i^0$$

Then, the power production of each generator belonging to $I$ is decreased proportionally:

$$\forall i \in Gen_I \ : \ PG_i^r = PG_i^{r-1} \frac{\sum_{i \in Load_I} PD_i^0}{\sum_{i \in Gen_I} PG_i^{r-1}} \tag{A8}$$

$$\forall j \in Load_I \ : \ PD_j^r = PD_j^0 \tag{A9}$$

In all the above equations, $PD_j^0$ denotes the power demand of node $j$ before the initial failure.

The major drawback of this strategy is related to the fact that it does not take into account the capacities of the surviving lines in the system in order to more effectively re-dispatch the power flows. Other more realistic re-dispatch strategies have been employed in previous works of literature. For example, in Ref. [5], the power supply and demand is adjusted by means of a linear optimization, whereas in Ref. [13], a controller proportionally sheds load or increases generation depending on the situation. On the other hand, with the strategy proposed here, simulations of cascading failures are much faster, thus allowing us to demonstrate the feasibility of statistical analysis and to perform optimization with reasonable computational efforts.

*Appendix A.3. Outage Model*

At step 5 of Algorithm 1, an outage model is required to identify the transmission lines which fail during iteration $r$. To this aim, we exploit the concept of effective power flow [12,34], where the transients of the power flows $x_l$ in the network lines after a disconnection event are modeled as follows:

$$x_l(t+1) = (1-\alpha)x_l(t) + \alpha F_l^r; \quad l \in E^r \tag{A10}$$

$$x_l(0) = H_l^{r-1} \tag{A11}$$

where the parameter $\alpha \in (0, 1]$ is called the thermal inertia of the grid, $F_l^r$ is the $l$-th component of the power flow vector $\boldsymbol{F}^r$ (step 3, Algorithm 1), $t$ is a discrete time index, and $H_l^{r-1}$ is the power flowing through line $l$ at the $(r-1)$-th failure event. The grid inertia $\alpha$ represents the "reactiveness" or "memory" of the grid with respect to any change in the power flows: if $\alpha = 1$, the system is memoryless and $x_l(t+1) = F_l^r$, thus implying an instantaneous power flow change after the $r$-th failure event, whereas $\alpha < 1$ implies a transient phase which becomes longer as $\alpha$ approaches 0 [12]. In order to identify the line disconnection to be considered in the cascading failure sequence, we first identify the set of the candidate line outages at round $r$ as the lines with a corresponding final power flow larger than the line capacity, i.e., $\widetilde{\boldsymbol{O}}^r = \left\{ l \in \boldsymbol{E}^r : F_l^r > C_l \right\}$ and the corresponding set of transient durations: $\widetilde{\boldsymbol{T}}^r = \left\{ t_l : l \in \widetilde{\boldsymbol{O}}^r, \min\{t : x_l(t) > C_l\} \right\}$. Then, the set of lines failed at the $r$-th failure event $\boldsymbol{O}^r$ and the corresponding transient duration $T^r$ are identified as follows:

$$T^r = \min\left\{ t : t \in \widetilde{\boldsymbol{T}}^r \right\} \tag{A12}$$

$$\boldsymbol{O}^r = \left\{ l : l \in \widetilde{\boldsymbol{O}}^r, t_l = T^r \right\} \tag{A13}$$

i.e., the candidate line outage with fastest transient. Note that (i) $t^r = \sum_{i=1}^{r} T^i$ is the time of occurrence of the $r$-th failure event (ii) $H_l^r = x_l(T^r)$, and (iii) $T^r$ is the scalar discrete time within which more than one failure event may, in principle, occur. As will be shown in the next section, a small value of $\alpha$ will be chosen, so that the unrealistic occurrence of multiple failure events in the same instant (possible in our model, due to its discrete time nature) becomes very rare; consequently, it will not affect the reproducibility and generality of the results.

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
