# Peer review of "Vulnerability Analysis of Power Transmission Grids Subject to Cascading Failures"

_electronics, doi:10.3390/electronics13050943_

Round 1

Reviewer 1 Report

Comments and Suggestions for Authors

The original contribution of the paper is the extension of the algorithm in [5] for sampling random power grid topologies with realistic electrical parameters. The main differences to [5] should therefore be highlighted more clearly in the paper (section 2).

A large extent of the paper is written around known topics (section 3). I recommend to significantly reduce this section and refer to literature instead. It is not necessarily needed for understanding the discussion of the results.

The discussion in section 4.1 is very long. What is missing is a way to generalize the conclusions drawn there to other types of base networks. It would be very helpful if you first present the different topological properties of the studied base networks (IEEE14 and IEEE118) in section 2 and then structure your discussion in section 4.1 around these properties. In this way, the relevance of the paper can be significantly increased.

The paper introduces a new metric (MS) for classifying the results of the cascading outage simulation. for Are there any further metrics than FO and FLS proposed in literature that could be used?

The performance of the genetic algorithm is not discussed sufficiently. For example, figure 9 shows that many randomly generated networks have a lower MS compared to the solution obtained by the genetic algorithm. This needs to be improved.

Author Response

We thank the reviewer for his comments. Below you can find our answers to your questions. The modifications are highlighted in yellow in the updated manuscript.

  1. The reviewer is probably referring to ref [25] instead of [5]. In this case, the difference is already highlighted in the Introduction: "To this aim, we first extend the algorithm of 25 in order to be able to sample also the power supply and demand locations and magnitudes".
  2. We agree. We have moved most of Section 3 to Appendix A, leaving only a general algorithm description in the main body text.
  3. We are not sure we fully understood the Reviewer's comment. In Section 2 we propose a general methodology that can be applied to any network topology (provided that we have sufficient computational power) and in Section 4 we present our case studies (i.e., IEEE-14 and IEEE-118) to demonstrate the effectiveness of the method. Thus, moving the presentation of the different topological properties of the studied base networks to Section 2 would appear slightly akward. Rather, we have split Section 4.1 into two sections. The new Section 4.1 introduced the case study, while Section 4.2 reports about the statistical analysis.
  4. There are many more possible metrics that can be used in this context. However, we believe that, among those available in literature, FO and FLS were the most intuitive and suitable to demonstrate our idea. Besides, the new metrics MS is an extension of FLS. In the interest of clarity, to avoid filling the manuscript with plots of tens of different metrics and performance indexes, we decided to stick to the most effective ones for our purposes.
  5. Section 4 already provides a thorough discussion of the optimization results. In fact, we state: "As already noted above, Figure 9 further shows that the maximum improvements on the single IEEE14 configuration due to the impedances and loads optimizations are smaller than the potential ones achievable by modifying the topology of the network". This implies that all the other combinations leading to lower MS values are actually associated to network topologies different from the IEEE-14 one, whereas those identified in the plot were just the results of an optimization of impedances and loads on the same IEEE-14 topology.

Reviewer 2 Report

Comments and Suggestions for Authors

Comments on the Quality of English Language

Quality of English language is good and easy to understand. However, there are several long sentenses that should be changes for better understanding.

Author Response

We thank the reviewer for his comments. Below you can find our answers to your questions. The modifications are highlighted in yellow in the updated manuscript.

1) We have shortened the most critical sentences in the manuscript.

2) We believe these are issues to be solved at an editing level (if the manuscript is accepted). Moreover, fixing such issues in the revised manuscript would not be beneficial, since the office who will be in charge of the editing process will for sure modify the paper layout. The bold font in Section 3.1 (which has been moved to Appendix A) was used for indicating matrices. In the interest of clarity, we stick to the choice of using bold for pointing out that some variables are matrices.

3) By "positive power generation" we just mean non-zero. The text has been accordingly modified.

4) As already stated right after Equation 3: "Alternatively, different strategies for more realistically assigning both the maximum generator power ("rho") and the line capacities ("epsilon") could in principle be followed, based, for example, on sampling values of both  and  from properly devised probability distributions, or on calculating them on the basis of physical laws and/or engineering practices. However, we believe this is out of the scope of the present work, which is mainly methodological, and that the assumptions made are sufficient to capture the average properties of cascading failures in power transmission networks". In any case, according to the previous work in [3], as also already stated in the manuscript, the choice has been made "to study the behavior of grids operating close to their limit, in the attempt of representing the conditions of old generation power grids overburdened by the continuously increasing power demand".

5) That is right, the grayscale colormap was not readable enough. A new colormap (copper) has been used and the new figures have been included in the revised manuscript.

6) Please, refer to the answer to question 4 of Reviewer 3.

Reviewer 3 Report

Comments and Suggestions for Authors

Dear Authors,

Your manuscript entitled: "Vulnerability Analysis of Power Transmission Grids Subject to Cascading Failures" touches important question of increased number of cases of cascading failures modeling in DC power transission networks. The article has a good potential however, it should be improved in several ways slightly. I also have a few questions.

1. The description of the axis in Figure 1 is not clear.

2. There is no explanation of the constant ρ in formula 2.

3. The axes in Figure 9 are not described. Please complete the axis descriptions.

4. Only one type of failure was assumed, related to the overload of subsequent lines. What about modeling the other failures?

5. What about the modeling of the AC networks?

6. Why the line losses are omitted in Your research? How does this simplification affect the obtained results?

7. What is the computational complexity of the algorithm?

8. Wouldn't modeling a specific network with a specific topology and analyzing the possible cascading failures based on it provide a better result?

Author Response

1) The x-axis indicates the number of overloaded lines observed during a cascading failure over the available number of lines, as described in the body text (see descriptions of the indicator FO at equation 17). The y-axis instead is the number of cascading failure simulations for which the indicator FO is computed.

2) Above equation 2, the body text states: "The maximum power that each generator can produce is assumed to be a fraction rho larger than nominal". Rho is then just a percentage of power increase above the nominal one.

3) Figure 9 was updated in the revised manuscript.

4) The scope of the work, as highlighted in Abstract and Introduction, is that of cascading failures due to overloads, which is the typical cause of large blackouts occurred in the past. Indeed the work could be extended and made more realistic in many ways, for example introducing more realistic failures, provided that they give rise to cascading phenomena. For example, a first extension could be that of introducing also bus (node) failures, which could fail more lines simultaneously, however we believe such an improvement could be worth one (or even more) independent publications. 

5) Please, see the answer to the previous question. Note also that AC power flows are significantly computationally heavier, so that it might become almost impossible to draw statistically valuable conclusions. As proposed by many other works of literature, we thus critically accepted the limitations of the DC power flow approach.

6) The main motivation lies in the fact that the introduction of line losses would have made simulating so many scenarios prohibitive with the available resources. By introducing them, we expect indeed some modifications of the results, but not of the general trends and behaviors identified by our research (e.g, we expect just shifts of the distributions estimated, but not significant changes in shape).

7) The computational complexity has not been evaluated in this work. However, most of the computational effort lies in the DC power flow solution, whose complexity is well known. It has then to be multiplied by the number of simulations and by the average number of cascading failures for each simulation (requiring power flow re-calculation).

8) We understand the Reviewer's point of view, but we do not agree. The whole aim of this work is that of drawing general conclusions on families (or classes) of similar networks, so that general guidelines for possible improvements (with respect to cascading failure robustness) can be provided. Studying an individual, specific network would have allowed to draw very specific and case-study-related conclusions.